# Recombinant Protein Vaccines Formulated with Enantio-Specific Cationic Lipid R-DOTAP Induce Protective Cellular and Antibody-Mediated Immune Responses in Mice

**DOI:** 10.3390/v15020432

**Published:** 2023-02-04

**Authors:** Siva K. Gandhapudi, Hua Shi, Martin R. Ward, John Peyton Bush, Margarita Avdiushko, Karuna Sundarapandiyan, Lauren V. Wood, Mania Dorrani, Afsheen Fatima, Joe Dervan, Frank Bedu-Addo, Greg Conn, Ted M. Ross, Jerold G. Woodward

**Affiliations:** 1Department of Microbiology, Immunology and Molecular Genetics, College of Medicine, University of Kentucky, Lexington, KY 40506, USA; 2PDS Biotechnology Corporation, Florham Park, NJ 07932, USA; 3Center for Influenza Vaccine Research for High Risk Populations, University of Georgia, Athens, GA 30602, USA; 4Center for Vaccines and Immunology and Department of Infectious Diseases, University of Georgia, Athens, GA 30602, USA

**Keywords:** R-DOTAP, influenza, SARS-CoV2, vaccine induced T cell immunity, immune stimulatory cationic lipids, vaccine adjuvants

## Abstract

Adjuvants are essential components of subunit vaccines added to enhance immune responses to antigens through immunomodulation. Very few adjuvants have been approved for human use by regulatory agencies due to safety concerns. Current subunit vaccine adjuvants approved for human use are very effective in promoting humoral immune responses but are less effective at promoting T-cell immunity. In this study, we evaluated a novel pure enantio-specific cationic lipid 1,2-dioleoyl-3-trimethylammonium-propane (R-DOTAP) as an immunomodulator for subunit vaccines capable of inducing both humoral- and cellular-mediated immunity. Using recombinant protein antigens derived from SARS-CoV2 spike or novel computationally optimized broadly reactive influenza antigen (COBRA) proteins, we demonstrated that R-DOTAP nanoparticles promoted strong cellular- and antibody-mediated immune responses in both monovalent and bivalent vaccines. R-DOTAP-based vaccines induced antigen-specific and polyfunctional CD8^+^ and CD4^+^ effector T cells and memory T cells, respectively. Antibody responses induced by R-DOTAP showed a balanced Th1/Th2 type immunity, neutralizing activity and protection of mice from challenge with live SARS-CoV2 or influenza viruses. R-DOTAP also facilitated significant dose sparing of the vaccine antigens. These studies demonstrate that R-DOTAP is an excellent immune stimulator for the production of next-generation subunit vaccines containing multiple recombinant proteins.

## 1. Introduction

Immunization remains one of the most effective public-health measures to combat viral infections. Increased safety requirements for approval by the FDA, combined with the complexities associated with certain viral infections, have invigorated the search for new generation peptide/recombinant protein-based prophylactic vaccines. However, most recombinant proteins/peptides are not immunogenic and only induce weak immune responses when administered. Hence, recombinant protein/peptide vaccines often require an adjuvant to stimulate or enhance the immune response to antigens [1,2]. 

There are a wide variety of adjuvants approved or under study for recombinant protein-based vaccines with different immune modulatory properties. Many adjuvants promote strong antibody responses, e.g., alum, or squalene, while others, e.g., monophosphoryl lipid A, STING, RIG- I, CpG or other TLR 7, 8 or 9 agonists promote stronger Th1 CD4 and CD8 T-cell responses [3,4]. However, even these distinctions are blurred somewhat because a single adjuvant can promote different responses through distinct pathways. For example, squalene-based adjuvants induce CD8 T-cell responses through a pathway distinct from its antibody-inducing property [5]. Thus, it is now appreciated that different adjuvants evoke distinct immunological signatures that render them more or less effective for distinct diseases, e.g., tuberculosis vs. influenza [6]. Subunit and mRNA vaccines against respiratory viruses, e.g., influenza or SARS-CoV2, are primarily focused on the induction of antibody responses to the envelope protein, e.g., hemagglutinin or spike and, for influenza, a neutralizing antibody titer is the primary accepted correlate of immunity [7,8]. While these vaccines also invoke T-cell responses, the degree to which these responses participate in anti-viral immunity and their duration is unclear. In contrast, natural infection also induces potent CD4 and CD8 T-cell responses which are critical in forming long-lasting immunity [9,10]. CD8 T-cell responses to internal proteins are also critical in situations where antigenic drift or shift prevents antibody recognition. Thus, it is now widely recognized that next generation universal vaccines against respiratory viral pathogens should induce strong neutralizing antibodies as well as strong CD8 T cell immunity [11,12]. 

Lipid nanoparticles are some of the most promising delivery vehicles for eukaryotic cells, and have been widely used since the late 1980s, including human gene-therapy clinical trials [13,14,15], for the delivery of nucleic acids into cells. The novel mRNA vaccine technologies used to fight the COVID 19 pandemic also use lipid nanoparticles as delivery vehicles to deliver mRNA into cells [16]. In recent years, cationic lipids have become attractive targets for delivering proteins or peptides for use in immunotherapy and vaccines [17,18,19,20]. Importantly, cationic lipid-mediated antigen uptake delivers proteins and peptides into the MHC class I and class II processing pathway [21,22,23,24]. Mechanistically, cationic lipid nanoparticles efficiently bind negatively charged cell membranes in a receptor-independent manner [25] and are rapidly internalized into endosomes in amounts exceeding receptor-mediated uptake [26,27]. Once in endosomes, cationic lipids fuse with the endosomal membrane, delivering some of their contents to the cytoplasm [28]. Thus, specific cationic lipids are ideally suited as non-viral vectors to deliver peptides, proteins and inactivated whole viruses intracellularly into the MHC class I and II pathways [29]. More recent studies investigating cationic lipids as delivery agents also identified that certain cationic lipids possess immune-stimulatory properties and can activate pathways essential for effective immune responses following vaccination [30]. However, because cationic lipids differ greatly in their immunostimulatory properties and mechanisms of action, these characteristics are not universally exhibited by all cationic lipids [31,32,33].

The enantio-specific cationic lipid 1,2-dioleoyl-3-trimethylammonium-propane (R-DOTAP) has been shown to be particularly robust at inducing CD8 T-cell responses to peptide-based vaccines [33,34]. In fact, a single vaccination with a vaccine containing an human papillomavirus 16 (HPV) peptide formulated with R-DOTAP resulted in complete regression of pre-existing tumors expressing HPV antigens [20,33]. Tumor regression was mediated by peptide-specific CD8 T-cell infiltration into the tumor. Further studies revealed that R-DOTAP promoted cellular uptake and cross-presentation of CD8 epitopes from long peptides and promoted the formation of polyfunctional CD8 T cells. R-DOTAP alone was shown to induce type I interferons in the draining lymph nodes (LN) and type I interferon was required for the R-DOTAP-mediated induction of antigen-specific CD8 T cells [33]. While it is well established that R-DOTAP facilitates robust CD8 T-cell responses to peptide-based vaccines, the ability of R-DOTAP to promote CD8 T-cell responses to larger recombinant proteins is less clear. While R-DOTAP was also shown to induce robust antibody responses to the model antigen, OVA [35], there have been no studies on the ability of R-DOTAP to promote antibody responses to vaccine-relevant recombinant proteins.

We now show that R-DOTAP promotes robust neutralizing antibody and T-cell responses to three different recombinant SARS-CoV2 spike protein variants, two different COBRA sequences encoding hemagglutinin proteins [36] and T cell responses to influenza nucleoprotein. Vaccination with a prototype SARS-CoV2/R-DOTAP vaccine or a multivalent H1/H3 COBRA/R-DOTAP vaccine resulted in protection from lethal challenge with SARS-CoV2 or influenza virus. Furthermore, vaccination with the bivalent COBRA/R-DOTAP vaccine resulted in a broad neutralizing antibody response to multiple H1 and H3 drift variants. These results suggest that R-DOTAP is a promising immune stimulant for the development of next-generation universal vaccines against respiratory viruses.

## 2. Results

### 2.1. Immunogenicity of Recombinant SARS-CoV2 Proteins Formulated with R-DOTAP

SARS-CoV2 enters host cells by binding to the host-cell ACE2 receptor with its spike protein. Because of its crucial role in host cell entry, spike protein is the primary target antigen used in prophylactic vaccines against SARS-CoV2. Immune responses interfering with viral attachment and fusion can prevent infection and provide protection to the host [37,38]. Hence, for this preclinical study, we chose the SARS-CoV2 spike protein as the target antigen for developing a prototype vaccine formulation. To assess the immunogenicity of the R-DOTAP containing the SARS-CoV2 vaccine formulation, we tested three variants of the spike protein as antigens. The first vaccine was prepared by formulating SARS-CoV2 spike receptor binding domain (RBD) with R-DOTAP nanoparticles. The RBD domain is the minimal protein domain that interacts with the human ACE2 receptor for entry [38]. The second vaccine was prepared by formulating recombinant SARS-CoV2 Spike S1 domain with R-DOTAP nanoparticles. The S1 protein domain (685aa) is the N-terminal region of the spike protein, which contains the receptor-binding domain required for ACE2 receptor binding. In the third formulation, we formulated full-length SARS-CoV2 spike protein with R-DOTAP nanoparticles. Due to their larger size, we hypothesized that the S1 domain and full-length spike as antigens would induce a strong neutralizing antibody response and strong T-cell-mediated immunity for long term protection.

To characterize T-cell responses induced by the R-DOTAP-containing vaccines, we immunized C57BL/6J mice with two doses of vaccine containing recombinant antigens formulated in sucrose buffer solution or with R-DOTAP nanoparticles. C57BL/6 (B6) mice were chosen due to the prior identification of H-2^b^-restricted CD8 and I-A^b^-restricted CD4 peptide epitopes within SARS-CoV2 spike [39,40]. Based on prior experience with numerous peptide antigens formulated with R-DOTAP, we have found that a primary and secondary vaccination spaced seven days apart, followed by assay seven days after the secondary vaccination yields strong reproducible T-cell responses in the spleen [33]. We measured antigen-specific T-cell responses by stimulating splenocytes from vaccinated mice with overlapping peptides (15 aa long) spanning the SARS-CoV2 S1 domain (1–685) or verified [39,40] H-2K^b^ binding CD8 T-cell epitope VL8 (SARS-CoV2 Spike (539–546): VNFNFNGL), or I-A^b^ binding CD4 T-cell epitope VT15 (SARS-CoV2 Spike (62–76): VTWFHAIHVSGTNGT) and measuring IFN-γ production using an ELISpot assay. Mice immunized with R-DOTAP containing vaccine formulations showed a highly significant increase in the number of antigen-specific IFN-γ producing T cells in spleens compared to the formulations containing antigen alone (Figure 1a–c). Among SARS-CoV2 spike protein variants, the R-DOTAP formulation consisting of spike S1 and RBD proteins showed more potent T-cell responses, as evidenced by the 20–50-fold increase compared to the antigen alone group, compared to the full-length spike protein with approximately 5-fold increase (Figure 1a). Both VL8 and VT15, verified CD4 and CD8 T-cell epitopes, induced significantly higher numbers of IFN-γ producing T cells in vaccinated groups containing R-DOTAP compared to antigen alone (Figure 1b), indicating that R-DOTAP-based vaccines promote strong CD4 and CD8 T-cell responses to recombinant protein antigens. Because the spike variants used were of different molecular weights, it is possible that the reduced T-cell response observed with full-length spike in Figure 1a was due to the fact that equivalent weights were used, resulting in a lower molar quantity of the larger proteins. Therefore, we vaccinated mice with formulations consisting of equimolar concentrations of the three spike variants and measured T-cell immune responses using ELISpot assay. We observed similar differences in T-cell responses between equal weights and equimolar concentrations of the spike antigen variants (Figure 1c) to both the spike S1 peptide pool and to the VL8 CD8 epitope. 

To evaluate antibody responses to the different spike variants, we performed ELISA assays on serum samples obtained on day 14 from vaccinated B6 mice. In contrast to the T-cell responses, we saw significantly higher levels of antibodies in mice vaccinated with spike full-length proteins compared to formulations containing RBD or spike S1 proteins (Figure 1d), indicating that immunogenicity of the spike protein variants differs between T-cell and antibody responses. 

To further characterize SARS-CoV2 RBD or spike S1 vaccines over an extended period of time, we vaccinated BALB/c mice subcutaneously on day 0 and day 7 and collected serum samples from vaccinated mice 15–62 days after initial vaccination to measure anti-RBD specific antibodies. In this study, BALB/c mice were used to assess the efficacy of R-DOTAP-based vaccines in diverse genetic backgrounds. Spike variant proteins formulated with R-DOTAP induced strong antibody responses within two weeks after vaccination (Figure 1e). As expected, antigen-only vaccine groups induced only weak antibody responses. Both antigen variants of spike protein induced comparable antibody titers specific to SARS-CoV2 RBD. The antibody titers induced by the R-DOTAP-based vaccine remained high even after 62 days following vaccination. 

### 2.2. Immunogenicity of Recombinant Influenza Protein Formulated with R-DOTAP Nanoparticles

We next assessed the ability of R-DOTAP to enhance the immunogenicity of recombinant influenza proteins. For this, we used influenza nucleoprotein (NP), or computationally optimized broadly reactive antigen (COBRA) hemagglutinins (HA) as vaccine antigens. For NP, we focused on only T-cell responses because NP is an internal influenza viral protein and the primary effective immune response to NP is mediated by T cells. Using our 14 day T-cell assay, we vaccinated B6 mice and assessed the T-cell immune response using a validated H2-D^b^-binding CD8 T-cell epitope [41] (NP366-74: ASNENMETM) and a validated I-A^b^ binding CD4 T-cell epitope [42] (NP-311-25: QVYSLIRPNENPAHK). R-DOTAP-based vaccines induced strong CD4^+^ and CD8^+^ T-cell responses to nucleoprotein epitopes compared to antigen-only vaccines (Figure 2a). 

For influenza hemagglutinin, both antibody and T-cell responses are critical for immunity. The COBRA HA proteins represent a novel approach toward a universal influenza vaccine because they induce broadly reactive antibodies that can neutralize multiple drift variants [36,43]. To better model a potential human vaccine designed to produce influenza-neutralizing antibodies and T-cell responses, we used intramuscular delivery and a vaccination schedule more commonly used for antibody production in mice. To determine whether COBRA HA proteins can induce T-cell responses when formulated with R-DOTAP, we prepared monovalent and bivalent vaccine (R-DOTAP-Y2NG2) containing recombinant COBRA sequences representing H1N1 hemagglutinin (Y2) and H3N2 hemagglutinin (NG2) formulated with R-DOTAP nanoparticles. We vaccinated BALB/cJ mice intramuscularly on day 0 and 21 and measured T-cell responses 7 days after the second dose using an IFN-γ ELISpot assay. Because the squalene-based adjuvant AddaVax is commonly used to study influenza vaccine responses, we also utilized that as a comparator. Because there are no defined T-cell epitopes within the COBRA HA proteins, we used whole COBRA proteins (Figure 2b) or overlapping peptides from A/California 07/2009 (H1N1) or A/New York/384/05(H3N2) hemagglutinins which share consensus with COBRA sequences and are relevant to the naturally circulating influenza viruses (Figure 2c,d). Mice vaccinated with monovalent R-DOTAP-Y2 or R-DOTAP-NG2 vaccine showed robust T-cell responses when whole COBRA protein antigens were used to stimulate in the ELISpot assay (Figure 2b). Furthermore, T-cell immune responses induced by R-DOTAP-adjuvanted formulations were significantly higher compared to AddaVax adjuvanted formulations. 

To test the potential immunogenicity of a bivalent H1/H3 vaccine, mice were vaccinated with R-DOTAP-Y2NG2 and T-cell responses assessed using overlapping peptides from naturally occurring H1N1 or H3N2 viruses in the ELISpot assay. We observed T-cell responses to multiple distinct peptide pools derived from A/California (H1N1) or A/New York (H3N2) indicating that T-cells raised against COBRA antigens can recognize and respond to multiple conserved T-cell epitopes presented by naturally occurring hemagglutinins from H1N1 and H3N2 strains of viruses (Figure 2c,d).

To assess antibody-mediated immune responses to COBRA proteins, serum samples from vaccinated mice were analyzed for antigen-specific antibody titers. Similar to SARS-CoV2 protein vaccines, we saw a significant enhancement of antibody titers with a single injection of the monovalent R-DOTAP-Y2 or R-DOTAP-NG2 vaccine (Figure 3a). A second dose further increased antibody titers and was significantly different from antigen-only groups. Vaccination of mice with a bivalent R-DOTAP-Y2NG2 vaccine also induced robust and durable antibody responses to Y2 and NG2 proteins (Figure 3b,c). When compared to AddaVax adjuvanted formulations, mice vaccinated with R-DOTAP-containing formulations showed higher antibody titers measured at day 35 and day 62 (Figure 3b,c). 

Previous studies with peptide antigens have shown that R-DOTAP is active over a wide dose range. This was confirmed with R-DOTAP-Y2 COBRA vaccines (Appendix A). Likewise, R-DOTAP has been shown to confer dose-sparing properties to vaccine antigens [33]. This was also confirmed with R-DOTAP-Y2 vaccines showing excellent antibody responses with as little as 0.35 µg of protein delivered (Appendix A).

Th1-type antibody-mediated immune responses play an important role in protection against viral infections [44,45]. To further assess the R-DOTAP-induced immune response, we measured antibody subclass titers following vaccination. We observed that R-DOTAP induced similar levels of IgG1 and IgG2a antibodies to Y2 and NG2 indicating a balanced Th1/Th2 response (Figure 3d,e).

We next characterized the functional ability of vaccine-elicited antibodies to block influenza virus interaction with sialic acids. We used the hemagglutination inhibition (HAI) assay, a widely accepted measurement predicting the influenza virus neutralizing antibodies to different strains of H1N1 and H3N2 viruses. Bivalent vaccines (R-DOTAP-Y2NG2) formulated with R-DOTAP showed significantly enhanced titers compared to antigen-only vaccines (Figure 4) to multiple drift variants of both H1N1 and H3N2 virus strains that share consensus with the COBRA-Y2, and COBRA-NG2, respectively. We observed robust HAI titers above the 1:40 threshold for all H1N1 viruses, even at the lowest dose (0.12 μg) of antigen tested. HAI titers against H3N2 drift variants were lower than HAI titers against H1N1 viruses, but still showed significant HAI titers above the 1:40 threshold for the 3μg dose. As expected, we saw very little neutralizing activity (HAI titer < 1:40) of antibodies induced with antigen-alone samples or from mock vaccinated mice. Together our results show that vaccination with bivalent COBRA antigens formulated with R-DOTAP can induce robust and balanced Th1 and Th2 antibody responses and can induce broadly cross-reactive antibodies that can neutralize several H1N1 and H3N2 drift variants.

R-DOTAP containing vaccines induced long-lasting memory T-cell responses and polyfunctional antigen-specific T cells to recombinant proteins.

Polyfunctional T cells capable of producing multiple cytokines play a key role in control of viral infection [46] and adjuvants vary in the degree to which they promote polyfunctional T cells [12]. We therefore evaluated the induction of polyfunctional T cells by the R-DOTAP-based vaccines. For these studies, we immunized mice with recombinant SARS-CoV2 spike S1 protein formulated with R-DOTAP and measured the frequency of gated (Appendix A) effector (CD44^+^CD62^−^) CD4^+^ and CD8^+^ T cells secreting IFN-γ, TNF-α, and IL-2 on day 28 using an intracellular cytokine staining assay (ICS). We found that the antigen-specific cytokine secreting CD8^+^ T cells increased significantly in R-DOTAP vaccine groups with a more diverse population of polyfunctional antigen-specific CD8+ T cells compared to antigen-only groups (Figure 5a,b). We also observed a more diverse population of polyfunctional antigen-specific CD4+ T cells in the R-DOTAP vaccine groups with the majority producing all three cytokines IFN-γ, TNF-α, and IL2 (Figure 5a,c). To determine if R-DOTAP can induce long-lasting memory T cells, we measured IFN-γ, and IL-4 producing T cells using ELISpot, 28 days after the second dose administered on day 28. We observed a significant increase in the memory T-cell frequencies in spleen producing Th1 (IFN-γ) and Th2 (IL-4) in the R-DOTAP vaccine groups (Figure 5d).

R-DOTAP can enhance immunogenicity of unadjuvanted seasonal influenza vaccines.

We evaluated if R-DOTAP can be used to enhance immunogenicity to existing human influenza vaccines. As a proof of concept, we used a trivalent, inactivated influenza vaccine, Fluzone (2011–12 formulation) consisting of a split virus hemagglutinin preparation derived from A/California/07/2009 X-179A (H1N1), A/Victoria/210/2009 X-187 (an A/Perth/16/2009-like virus) (H3N2) and B/Brisbane/60/2008. We prepared vaccine formulations by mixing a 1:1 ratio of R-DOTAP nanoparticles and varying doses of Fluzone. C57BL/6J mice were vaccinated (0.1 ml/dose) on day 0 and day 21 and blood was obtained for HAI titers on day 35. We observed that formulating Fluzone with R-DOTAP significantly enhanced the HAI titers to all viral strains (Figure 6) in vaccinated mice compared to Fluzone-only vaccinated groups. Importantly, we saw a significant dose sparing effect in R-DOTAP vaccine groups.

Recombinant proteins formulated with R-DOTAP protected mice from challenge with SARS-CoV2, or influenza virus.

To determine the protective efficacy of R-DOTAP-based vaccines against SARS-CoV-2, we vaccinated K18hACE2 transgenic mice with vaccines containing SARS-CoV2 spike S1 protein with or without R-DOTAP, or with R-DOTAP alone. K18hACE2 transgenic mice were used because they are permissive for infection with SARSCoV2. Mice were then challenged with 2.5 × 10^4^ pfu/dose of the SARS-CoV2 virus. Mice vaccinated with spike S1 alone had low but significant titers of RBD-specific antibodies. As expected, formulations containing R-DOTAP had significantly higher titer compared to antigen alone groups (Figure 7a). Mice vaccinated with R-DOTAP alone rapidly lost body weight and 100% succumbed to SARS-CoV2 challenge (Figure 7b,c). Mice vaccinated with spike S1 alone also lost weight and showed 50% mortality. In contrast, 100% of the mice vaccinated with the R-DOTAP–based spike S1 survived the challenge without losing body weight indicating that these mice were completely protected against viral challenge (Figure 7b,c).

We next tested the protective efficacy of bivalent influenza vaccine R-DOTAP-Y2NG2 in mice. For this, we vaccinated DBA/2J mice with formulations containing varying doses of COBRA antigens with or without R-DOTAP in a two-dose regimen followed by challenge with A/Brisbane/2/2018 (H1N1) (3.6 × 10^6^ pfu/dose) 28 days post vaccination. DBA/2J mice were chosen because of an increased susceptibility to influenza virus infection over other strains. Weight loss was measured as the predictor of protection from challenge. Both unvaccinated mice and mice vaccinated with antigen-only (3 µg/HA) vaccine formulations rapidly lost body weight with 0 and 33% survival, respectively, within 7 days (Figure 7d,e). In contrast, we saw less than 5% weight loss and 100% survival in all groups vaccinated with the bivalent vaccine formulated with R-DOTAP (Figure 7d,e). Both low dose (0.12 ug/HA) and high dose (3 ug/HA) COBRA antigen formulated with R-DOTAP provided complete protection indicating a significant dose-sparing effect (Figure 7d). We also evaluated the viral clearance in the lungs following challenge. Vaccine formulations containing R-DOTAP completely cleared influenza virus in the lungs within 3 days (Figure 7f) and no virus was detected by day 6 (Figure 7g). In contrast, mice receiving antigen-only and unvaccinated mice showed significant viral load both at day 3 and day 6. Together, these studies demonstrated that recombinant protein vaccines containing R-DOTAP as immune modulator can induce immune responses capable of protecting mice from viral infections. Furthermore, these studies suggest that intramuscular vaccination with R-DOTAP-based vaccines induce mucosal immune responses sufficient to protect mice against challenge with two different respiratory viruses, SARSCoV-2 and influenza.

## 3. Discussion

Cationic lipids are excellent delivery vehicles for transporting nucleic acids and protein/peptides into cells and have been widely used in human drug delivery. However, most cationic lipids are inert and do not activate immunological signals necessary for effective immune response to vaccine antigens. In an earlier study, we showed that R-DOTAP is a highly effective peptide antigen-delivery vector that is effective in promoting multi-epitope peptide antigen-processing and presentation into the MHC Class I and Class II presentation pathways and delivers the necessary immunological signals required for robust T-cell responses [33]. Furthermore, R-DOTAP alone was shown to induce type I interferons in the draining LN [33], and promote maturation of dendritic cells and induction of chemokines and cytokines [18,47]. Here, we show that these properties also apply to a variety of recombinant protein antigens. In addition, very few studies have evaluated the ability of R-DOTAP to promote antibody responses to recombinant proteins. The data presented here demonstrate that R-DOTAP promotes robust antibody and T-cell responses to a variety of respiratory virus proteins capable of providing neutralizing activity and protection from viral challenge.

Earlier studies demonstrated that the interaction of the antigen with the nanoparticle either by encapsulation or surface interaction is essential for the antigen delivery and cross-presentation by cationic lipids [22]. Both TEM pictures (Appendix A) and physical characterization data (Appendix A and Appendix A) showed that R-DOTAP formed uniform smooth surface spherical structures ranging around 150 nm size in the sucrose buffer. Adding COBRA proteins to the R-DOTAP produced particles with a rough exterior surface suggesting protein interactions. Mixing COBRA HA antigens in either a monovalent or bivalent formulation with the nanoparticles did not significantly alter the size and polydispersity of the formulation (Appendix A). We saw a slight reduction in the zeta potential of the nanoparticles following antigen addition, but the nanoparticles still retained a net positive charge characteristic of stable particles. While deeper investigations are needed and are ongoing to quantitate antigen–lipid associations, changes in the physical characteristics of nanoparticles and potent immune responses in the presence of R-DOTAP nanoparticles together indicate an association of R-DOTAP nanoparticles with the protein antigens.

In this study, we evaluated the ability of R-DOTAP to enhance both humoral and cellular immune responses to large protein antigens using two prototype vaccine formulations containing antigens derived from two respiratory viruses, SARS-CoV2 or iInfluenza. By formulating R-DOTAP with the COBRA influenza antigens we also evaluated the potential for the R-DOTAP-COBRA vaccine to provide an effective universal flu vaccine by inducing a broadly protective immune response that may neutralize multiple strains of the influenza virus (Figure 4). We demonstrated that co-formulating SARS-CoV2 or iInfluenza derived viral protein antigens with R-DOTAP significantly enhanced vaccine immunogenicity, generated robust antigen-specific cellular and antibody-mediated immune responses in mice, mediated significant antigen dose-sparing and protected vaccinated mice from SARS-CoV2 and influenza in the virus challenge.

To date, adjuvants have been used to enhance the immunogenicity of recombinant protein vaccines primarily by providing the immune-modulation needed for protective humoral immune responses. Most adjuvant systems approved for human use have demonstrated a sub-optimal ability to promote antigen cross presentation and subsequent induction of durable CD8+ T-cells in addition to antibody responses [1]. Our results demonstrate that single component R-DOTAP nanoparticles can perform multiple tasks essential to generating broad and durable protective immune responses. We have previously shown that R-DOTAP promotes rapid protein antigen-uptake and cross-presentation on MHC class I by antigen-presenting cells [33]. In addition, we demonstrated that R-DOTAP stimulates type I IFN production, likely through the activation of endosomal TLR 7 and/or 9. Furthermore, the ability of R-DOTAP to promote CD8 T-cell responses was dependent on type I IFN and MyD88, but not TRIF or STING.

There is ample evidence that suggests CD8 T cells play a crucial role in long-term protection against highly mutating viruses such as SARS-CoV2 and influenza [11,12]. While most approved adjuvants for recombinant proteins effectively induce humoral and Th1-type immune responses, very few if any can induce robust and clinically effective cytotoxic CD8 T-cell immune responses in humans. Studies have already demonstrated that an R-DOTAP-based multi-epitope peptide immunotherapy induced antigen-specific cytotoxic CD8+ T cells producing IFNγ and granzyme B in human phase I trials (NCT02065973 and pre-clinical studies) [33]. The current study demonstrates a similar ability of R-DOTAP to generate CD8 T cells to internal epitopes of large recombinant protein antigens. These T cells were polyfunctional with an effector phenotype. They were capable of producing multiple cytotoxic cytokines and persisted in vaccinated mice 28 days after a second vaccine, indicating the establishment of T-cell memory responses. It is highly probable that this T-cell-inducing property of R-DOTAP in recombinant protein vaccines is due to both effective antigen delivery, cross presentation and type I IFN production as we have shown for peptide antigens [33].

Adjuvants can be broadly categorized into Th1, Th2 and mixed Th1/Th2 types based on the cytokines and antibody subclasses induced by the vaccine. Th1 type adjuvants, for example, show skewing towards IFN-γ production and IgG2a/c antibody subtypes in mice. Th2 adjuvants stimulate more IL-4 production resulting in skewing towards IgG1 antibody subtypes in mice [48]. We observed that R-DOTAP-containing vaccines induced both IFN-γ- and IL-4-producing T cells (Figure 5b) and generated both IgG1 and IgG2a antibody subtypes. We did not see any strong skewing towards either the Th1 or Th2 subtype. Thus, from an antibody perspective, these results indicate that R-DOTAP induces a balanced Th1/Th2 type immunity optimal for effective vaccine-induced antibody responses. However, among antigen-specific Th1 and CD8 T cells, R-DOTAP promotes the development of high levels of polyfunctional cytokine secreting cells shown to be optimal in promoting viral clearance [12].

With the continuing emergence of new SARS-CoV2 variants that can evade vaccine-induced immunity, there is now an effort to produce next-generation “universal” vaccines that will show broader protection. Our results showing that R-DOTAP promotes T-cell responses to multiple CD4 and CD8 epitopes within the spike protein suggests the potential for a broader SARS-CoV2 vaccine due to more conserved T-cell epitopes within spike. In addition, our results suggest the possibility of including internal SARS-CoV2 proteins, such as nucleocapsid, along with the spike protein and R-DOTAP to produce a multivalent vaccine that would protect against multiple SARS-CoV2 variants.

Development of a universal influenza vaccine that can provide heterosubtypic immunity and protection against multiple clades has been a long-term goal to protect against influenza infections. There are several strategies [49] under investigation to achieve this goal, including the use of antigens from multiple subtypes to increase the coverage, targeting multiple proteins (e.g., inclusion of HA and non-HA proteins) as antigens, target conserved antigenic regions of influenza antigens, use of chimeric proteins containing stem and stalk HA from different subtypes [50] and use of consensus-based approaches such as COBRA sequences [43] comprising multiple known mutations in the hemagglutinin or neuraminidase. In addition, vaccine technologies that induce both CD8 T cell- and antibody-mediated immunity are also actively being sought to achieve heterosubtypic protection against influenza. In this study, we have demonstrated, using a prototype vaccine based on the R-DOTAP platform and COBRA H1N1- and H3N2-derived HA or nucleoprotein antigen, that such a vaccine is capable of inducing antigen-specific T-cell responses, neutralizing antibodies against multiple clades of H1N1, and H3N2 strains, and protected mice against challenge with a lethal H1N1 strain. Hence, vaccine formulations containing R-DOTAP and COBRA HA, NA sequences and nucleoprotein from influenza presents strong potential to further the goal of developing a safe and effective universal influenza vaccine.

In summary, these studies demonstrate that protein subunit vaccines against SARS-CoV2 and influenza that are based on the R-DOTAP platform can induce broadly protective cellular and humoral immune responses and provide a significant dose-sparing effect on the antigen. The heterogeneity of recombinant proteins capable of being formulated and administered with R-DOTAP coupled with the ability to produce multivalent vaccines suggest that R-DOTAP is an excellent candidate for use in a variety of prophylactic vaccines against infectious disease. Furthermore, we have demonstrated that R-DOTAP is highly effective in enhancing the potency of the currently licensed seasonal influenza vaccine, further expanding the potential utility of R-DOTAP. The safety and efficacy profiles of R-DOTAP have been successfully established in human clinical trials; thus, paving the way for future trials of R-DOTAP-based universal influenza and SARS-CoV2 vaccines.

## 4. Materials and Methods

### 4.1. Animals and Viruses

Six to twenty-week-old C57BL/6J mice (B6 mice), BALB/cJ, K18-hACE2 mice (B6.Cg-Tg(K18-ACE2)2Prlmn/J), and DBA/2J, mice were obtained from Jackson Laboratories (Bar Harbor, M.E.). All animals were housed in specific-pathogen-free conditions at the Division of Laboratory Animal Resources (DLAR), University of Kentucky Medical Center, or in the animal research center at the University of Georgia. SARS-CoV2 cChallenge studies were conducted at the animal research centers at the University of Georgia under biosafety 3 containment. All animal protocols were reviewed and approved by the University of Kentucky (2019–3226) or The University of Georgia Institutional Animal Care and Use Committee following the National Institutes of Health Animal care guidelines (A2020 02-024-Y1-A5, A2018 06-018-Y3-A16). All studies were carried out in compliance with ARRIVE guidelines. For influenza challenge studies (A/Brisbane/02/2018(Bris/18) was used. For HAI assays (A/California/07/2009 (Cal/09), A/Guangdong-Maonan/SWL1536/2019(GD19), A/Singapore/IFNIMH-16-00192016 (Singapore/16), A/Hong Kong/4801/2014 (Hong Kong/14), A/Victoria/210/2009 X-187 (an A/Perth/16/2009-like virus) (H3N2) and B/Brisbane/60/2008 were used. Influenza viruses were obtained from International Influenza Resources (IRR) [36]. For SARS-CoV2 challenge studies, 2019n-CoV/USA_WA1/2019 isolate was obtained from BEI resources.

### 4.2. Reagents and Antibodies

cGMP grade R-DOTAP (1, 2-dioleoyl-3-trimethyl ammonium-propane) was provided by Merck & Cie (Shaffhausen, Switzerland). Evonik (Vancouver, Canada) produced cGMP grade R-DOTAP liposomal nanoparticles according to protocols described previously [33]. SARS-CoV2 recombinant vaccine antigens: SARS-CoV2 spike RBD domain protein (S-RBD) (Cat# Z03479), SARS-CoV2 spike S1 (S1 spike) (cat# Z03485) and SARS-CoV2 Spike protein (Spike) (cat # Z03481) were purchased from Genscript Piscataway, New Jersey, USA. SARS-CoV2 Spike RBD domain protein used for ELISA assays were produced at the protein core facility at University of Kentucky using an expression vector provided by Dr. Florian Krammer at Icahn School of Medicine at Mount Sinai. Influenza COBRA antigens: COBRA HA-Y2 (COBRA-Y2) (H1N1) and COBRA-HA-NG2 (COBRA-NG2) (H3N2), were synthesized at the University of Georgia Center for Vaccines and Immunology Research center [36]. Influenza nucleoprotein from A/Puerto Rico/8/34/Mount Sinai) (cat # 11675-V08B) was obtained from Sino Biologicals US Inc. Fluzone vaccine (2011–12) formulation was obtained from the University of Kentucky HealthCare pharmacy. Research grade synthetic peptide antigens: VL8 (spike (539–546) VNFNFNGL) and VT15 (spike (62–76) VTWFHAIHVSGTNGT), were synthesized and purified to >95% purity by GenScript, Piscataway, New Jersey, USA. Overlapping peptide pools from SARS-CoV2 spike S1 protein (S1 pool), were purchased from Miltenyi Biotec (Peptivator S1 pool, Miltenyi Biotec. Cat#130-127-041). Overlapping peptide pools from influenza A/New York/383/2005 (H3N2) hemagglutinin protein (cat# NR-2603) and overlapping peptide pools from influenza A/California/07/2009 (H1N1) pdm09 (Cat# NR-19244) were obtained from BEI Resources. Fluorochrome-conjugated mouse monoclonal anti-mouse CD3 (Clone: 145-2c11), CD4 (Clone: GK1.5), CD8 (Clone: YTS165.7.7), CD44 (clone: IM7), CD62L (clone: MED-14), IFNγ (Clone: XMG1.2), TNFα (Clone: MP6-XT22) and IL-2 (Clone: JES6.5H4), were purchased from BioLegend, San Diego, CA, USA.

### 4.3. Preparation of R-DOTAP Nanoparticles and Vaccine Formulations

cGMP grade R-DOTAP liposomal nanoparticles were produced by Evonik (Vancouver, Canada). Briefly, R-DOTAP solid powder was added to a 5 L Bellco glass vessel with a vaned Teflon overhead impeller containing 280 mM low endotoxin sucrose in purified water. The mixture was stirred for two hours until all the R-DOTAP powder had been hydrated and had formed a uniform nanoparticle suspension. The R-DOTAP nanoparticle suspension was sequentially extruded five times over six stacked 0.2 μm porosity Whatman polycarbonate Nucleopore 142 mm diameter membrane filters using a Lipex high pressure extrusion system to obtain uniform-sized liposomal nanoparticles with a target size of 100–200 nm. The extruded nanoparticles were subjected to clarifying filtration over a 0.2 μm membrane, then sterile filtered over dual 0.2 μm sterile filter cartridges. The bulk sterile nanoparticles were vialed into 5 mL volume sterile endotoxin-free borosilicate glass vials at a fill volume of 1.2 mL/vial and sealed with butyl rubber snap-cap stoppers. The sterile-vialed nanoparticle product suspension at a final concentration of 6.0 mg/mL was stored at −80 °C. For making vaccine formulations, concentrated antigens dissolved in PBS buffer were diluted to the desired concentration in 280 mM sucrose. Prior to vaccination, the vaccine components were brought to ambient temperature and antigen components were then mixed at a 1:1 ratio with the R-DOTAP nanoparticles using a pipette to form a uniform suspension.

### 4.4. Physical Characterization of Vaccine Formulation

Particle size, polydispersity and zeta potential of R-DOTAP liposomes and vaccine formulations were measured at 23 °C using Zetasizer nano (Malvern Instruments Ltd., UK) equipped with a 4 mW 632.8 nm laser set at 90° angle. Dynamic light scattering was used to determine the fluctuations in dispersed light intensity. Distribution analysis and cumulants analysis to measure Z average and polydispersity was performed according to Malvern protocols and using apparatus software. A representative particle size distribution is shown in Appendix A. Average particle size, polydispersity and zeta potential measurements are shown in Appendix A. Electron micrographs of R-DOTAP in 280 mM sucrose and vaccine formulation (Y2.NG2.RDOTAP) in 280 mM sucrose were captured using Philips CM12 electron microscope equipped with AMT-XR11 digital camera.

### 4.5. Enzyme-Linked Immunosorbent Assay (ELISA)

Blood was collected into B.D. microtainer serum-separator tubes through tail vein bleed or cardiac puncture of euthanized mice. Isolated serum samples were stored at −80 °C until analysis. For antibody titer measurement, 96 well plates (Immulon 4HBX; ThermoFisher Scientific) were coated overnight at 4 °C with 50 µL/well recombinant RBD protein, COBRA-NG2 or COBRA-Y2 protein at 2 µg/mL concentration. After antigen coating, the wells were blocked for 1–2 h at room temperature with 200 µL/well of PBS-T buffer containing 3% non-fat dry milk (American Bio, Cat# AB10109) and 0.1% Tween-20. Following blocking, the buffer was replaced with serum samples diluted in PBS-T buffer containing 1% non-fat dry milk. After 2 h incubation at R.T., the serum was removed and wells were washed four times with PBS-T buffer. For detecting protein antibodies, wells were incubated for 1 h with 100 µL of PBS-T buffer containing 1% non-fat dry milk and HRP-conjugated anti-mouse IgG (1: 5000) (cat#115-035-003 Jackson ImmunoResearch), anti-mouse IgG1(1:5000) (cat#115-035-205; Jackson ImmunoResearch), anti-mouse IgG2c (1:5000) (Cat#115-035-206; Jackson ImmunoResearch) or anti-mouse IgG2a (1:5000) (Cat#115-035-206; Jackson ImmunoResearch). Wells were then washed four times, and 100 µL of SIGMAFAST OPD (o-phenylenediamine dihydrochloride; Sigma Cat# P9187) or 1-Step Ultra TMB chromogenic substrate was added to each well. After 10 min of incubation, the reaction was stopped by adding 50 µL of 3 M HCL, and absorbance measured at 490 nm (OD490) for OPD substrate or at 450 nm (OD450) for TMB substrate using a SpectraMax M5 microplate reader.

### 4.6. Enzyme-Linked Immunospot Assay (ELISpot)

Processed spleen cells, 2.5 × 10^5^, were stimulated for 18–24 h at 37 °C with T-cell epitope peptides (10 µg/mL), peptide pools (1 µg/mL per peptide) or recombinant proteins (10 µg/mL, or no peptides (control) in a 96 well plate pre-coated with the mouse IFNγ or IL-4 capture antibody (Mabtech, Inc., Cincinnati, OH, USA). After stimulation, wells were washed with PBS and incubated with biotin-conjugated anti-IFNγ or IL-4 antibody followed by streptavidin-HRP antibody. To visualize the antigen-specific IFNγ- or IL-4+producing cells, wells were incubated for 6 min with TMB substrate, washed with water, and air dried. Spots were scanned and counted using CTL ImmunoSpot Analyzer and ImmunoSpot Ver.6 software (Cellular Technology Limited, Cleveland, OH, USA). Spot counts were summarized as median values from triplicate samples. Each sample had unstimulated and PMA/Ionomycin control wells to detect background or as a positive control.

### 4.7. Intracellular Cytokine and Cell Surface Staining

For intracellular protein analysis, single cell suspensions were produced after thawing frozen spleen cells and rested overnight in cell-culture media (cRPMI media). Following resting, cells were stimulated with indicated stimulatory antigenic peptide for 6 h at 37 °C in cRPMI media supplemented with purified anti-mouse CD28 (2 µg/mL), protein transport inhibitor Brefeldin A (5 μg/mL) and monensin (2.0 μM). Following stimulation, cells were washed with FACS buffer and stained with fluorochrome conjugated anti-mouse CD3 (clone; 145-2C11), CD4 (clone; GK1.5), CD8 (clone:YTS156.7.7), CD44 (clone:IM7), and CD62L (clone:MEL-14) antibodies. The cells were then washed, fixed and permeabilized using a fixation/permeabilization kit (B.D. Bioscience) and stained with fluorochrome conjugated anti-mouse IFNγ (clone: XMG1.2), TNFα (clone: MP6-XT22), and IL-2 (clone: JES6.5H4). Cells were washed with FACS buffer after intracellular staining and analyzed immediately using flow cytometry and gating strategy detailed in Appendix A.

### 4.8. Mouse Vaccination

Mice were anesthetized using isoflurane for all injections and implants. The injection site was shaved and cleaned with 70% ethanol prior to subcutaneous or intramuscular injection of a formulation. For subcutaneous (S.C.) vaccination, a 100 μL dose was delivered on a single flank of the hind limb and for intramuscular (I.M.) vaccination, a 50 μL dose was delivered into the thigh muscle of the hind limb. For making R-DOTAP-based vaccine formulations, R-DOTAP nanoparticles (4–6 mg/mL) in 280 mM sucrose buffer were mixed 1:1 with indicated concentrations of recombinant proteins resuspended in 280 mM sucrose buffer. For antigen-only vaccine formulations, recombinant protein was resuspended at the desired concentration in 280 mM sucrose buffer. All vaccination regimens consisted of two doses delivered at 1–4 week intervals.

### 4.9. Hemagglutination Inhibition Assay

The hemagglutination inhibition (HAI) assay was used to assess functional antibodies to the HA that are able to inhibit agglutination of guinea pig erythrocytes for H3N2 viruses, and turkey erythrocytes for H1N1 viruses. The protocols were adapted from the World Health Organization (WHO) laboratory influenza surveillance manual [51]. Guinea pig red blood cells are frequently used to characterize contemporary A(H3N2) influenza strains that have developed a preferential binding to alpha (2,6)-linked sialic acid receptors. To inactivate nonspecific inhibitors, sera samples were treated with receptor-destroying enzyme (RDE) (Denka Seiken, Co., Chuo, Japan) prior to being tested. Briefly, three parts of RDE was added to one part of sera and incubated overnight at 37 °C. RDE was inactivated by incubation at 56 °C for 30 min.

RDE-treated sera were diluted in a series of two-fold serial dilutions in v-bottom microtiter plates. An equal volume of each A(H3N2) virus, adjusted to approximately 8 hemagglutination units (HAU)/50 μL in the presence of 20 nM Oseltamivir carboxylate, was added to each well. The plates were covered and incubated at room temperature for 30 min, and then 0.75% guinea pig erythrocytes (Lampire Biologicals, Pipersville, PA, USA) in PBS were added. Prior to use, the red blood cells (RBCs) were washed twice with PBS, stored at 4 °C, and used within 24 h of preparation. The plates were mixed by gentle agitation, covered, and the RBCs were allowed to settle for 1 h at room temperature. The HAI titer was determined by the reciprocal dilution of the last well that contained non-agglutinated red blood cells (RBCs). Positive and negative serum controls were included for each plate.

In separate assays, RDE-treated sera were diluted in a series of two-fold serial dilutions in v-bottom microtiter plates. An equal volume of each influenza virus, adjusted to approximately 8 hemagglutination units (HAU)/50 μL was added to each well. The plates were covered and incubated at room temperature for 20 min with erythrocytes (Lampire Biologicals, Pipersville, PA, USA) and phosphate-buffered saline (PBS) added. Prior to use, the RBCs were washed twice with PBS, stored at 4 °C and used within 24 h of preparation. The plates were mixed by gentle agitation, covered, and the RBCs were allowed to settle for 30 min at room temperature. The HAI titer was determined by the reciprocal dilution of the last well that contained non-agglutinated RBCs. Positive and negative serum controls were included for each plate.

All mice were negative (HAI ≤ 1:10) for pre-existing antibodies to human influenza viruses prior to infection or vaccination, and for this study sero-protection was defined as HAI titer > 1:40 and seroconversion as a 4-fold increase in titer compared to baseline, as per the WHO and European Committee for Medicinal Products to evaluate influenza vaccines [52]. All mice were naïve and seronegative at the time of vaccination, and thus sero-conversion and sero-protection rates are interchangeable for this study.

### 4.10. Mouse Challenge Experiments

For influenza virus challenge studies, the DBA/2J mice (female, 7 to 9 weeks old) were immunized with indicated vaccine formulations intramuscularly on day 0 and day 28. On day 56, they were challenged with the H1N1 A/Brisbane/02/2018 (Bris/18) influenza virus at an 10X LD_50_ dose of 3.6 × 10^6^ pfu/mouse intranasally with the volume of 50 μL. Mock-vaccinated animals were inoculated intranasally with 50 uL of PBS. For the SARS-CoV2 challenge studies, the K18-ACE2 transgenic mice were immunized intramuscularly with indicated doses of vaccine formulations on day 0 and day 28. On day 56, immunized mice were challenged intranasally with the SARS-CoV2 (2019n-Cov/USA_WA1/2019 isolate) virus at an 10X LD_50_ dose of 2.5 × 10^6^ pfu/mouse in a BSL3 biocontainment facility. All animals challenged with live virus were monitored twice daily, morning and evening, for weight loss and clinical signs (labored breathing, lethargy, hunched back, ruffled fur, failure to respond to stimuli, and severe respiratory distress), for up to 14 days post infection. The body weight was tightly monitored until 14 days post infection. Mice were humanely euthanized once they lost 20% of their original body weight or they reached clinical endpoints. Lungs in influenza-challenged mice were collected from three pre-selected mice in each group on day 3 and day 6 post-infection for viral titer detection [36]. Briefly, frozen lungs were processed and clarified supernatants containing virus were added to MDCK cells (Sigma-Aldrich) at 90% confluence and incubated for 1 h. following this step, cells were washed and supplemented with media containing 1.6% agarose (Thermo Fisher). After 72 h incubation at 37 °C, plates were processed, dried and viral plaques were enumerated plaque forming units per gram of lung tissue.

### 4.11. Equipment, Software and Statistical Analysis

Flow cytometry was performed using B.D. Symphony A3 flow cytometer equipped with BD FACSDiva software. All flow data were analyzed using FlowJo version 10.0 software. Statistical analyses for all other studies were performed using GraphPad Prism 9.0 software and comparing means by a simple student’s *t*-test or by ANOVA with Tukey multiple comparison correction. Mantel–Cox test was used for survival curves.

## Figures and Tables

**Figure 1 viruses-15-00432-f001:**
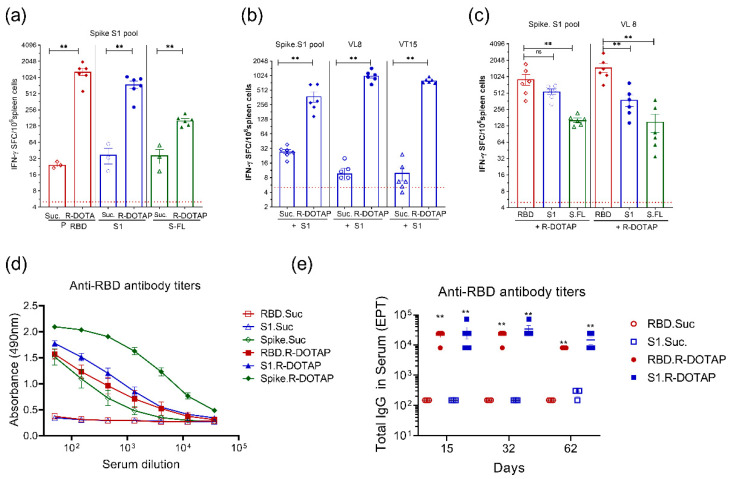
Recombinant SARS-CoV2 spike protein formulated with R-DOTAP nanoparticles induced potent T-cell and antibody-mediated immune responses. Groups of C57BL6/J mice (*n* = 5) (**a**–**d**) were subcutaneously immunized on day 0 and day 7 with 0.1 mL vaccine formulations containing SARS-CoV2-derived recombinant spike RBD domain (RBD), Spike S1 domain (S1), or Spike full-length protein (spike) formulated with R-DOTAP or sucrose buffer (Suc). Each mouse received a vaccine formulation containing 10 μg protein/dose (**a**,**b**,**d**) (equal weight basis) or estimated 750 nM protein/dose (equal molar concentration) (**c**) of indicated recombinant protein antigens. Splenocytes (**a**–**c**) isolated 7 days after second vaccine were stimulated overnight with pools of overlapping peptides (1 µg/peptide/mL) (S1pool) derived from SARS-CoV2 Spike S1 domain or VL8 CD8 T-cell epitope (10 µg/mL) or VT15 CD4 T-cell epitope (10 µg/mL) and IFN-γ producing cells in spleens from vaccinated mice were measured using an ELISpot assay. Data represent mean ± SEM spot forming cells (SFC) in each mouse. Dotted lines in each figure represent average SFC count in the background no peptide wells. Serum samples from the same group of mice shown in (**a**) were obtained on day 14 (**d**) and assayed for anti-RBD total IgG by ELISA. Serum samples from BALB/c mice (**e**) vaccinated on days 0 and 7 were obtained on days 15, 32, or 62 and assayed for anti-RBD total IgG antibody by ELISA. Data (**d**) represent mean ± SEM of relative absorbance (OD) at 490 nm. Data (**e**) represent mean ± SEM end point titer (EPT) from each mouse. Comparisons between sucrose alone or R-DOTAP groups (**a**–**c**,**e**) was performed using Student’s *t*-test (unpaired-two tailed). ** *p* ≤ 0.05 ^ns^
*p* ≥ 0.05. Data shown are representative of three independent repeat studies.

**Figure 2 viruses-15-00432-f002:**
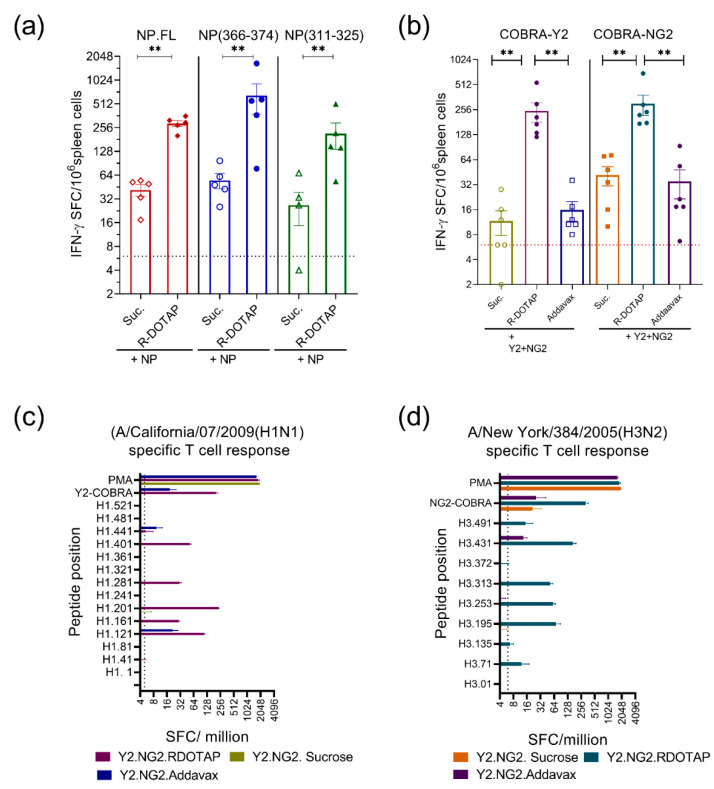
Recombinant influenza protein formulated with R-DOTAP nanoparticles induced potent antigen-specific T-cell immune responses. Groups (**a**) of C57BL/6J (*n* = 5) mice were subcutaneously immunized on day 0 and day 7 with two doses of vaccine formulations containing recombinant influenza NP (PR8) (5 µg/dose) formulated with R-DOTAP nanoparticles (R-DOTAP) or sucrose buffer (sucrose). Splenocytes isolated 7 days after second vaccine were stimulated overnight with NP (NP.FL), NP _(366–74)_ CD8 T-cell epitope or NP _(311–25)_ CD4 T-cell epitope and IFN-γ producing cells were measured using an ELISpot assay. Groups (**b**–**d**) of BALB/cJ (*n* = 6) mice were intramuscularly injected on day 0 and day 21 with bivalent vaccine formulations containing COBRA-NG2 (H3) (3 µg/dose) and COBRA-Y2 (H1) (3 µg/dose) formulated with R-DOTAP nanoparticles or sucrose buffer (sucrose). Splenocytes isolated 7 days after second vaccine were stimulated overnight with recombinant COBRA-Y2 or COBRA-NG2 proteins (**b**) or with pools of 8–10 overlapping peptides derived from A/California/07/2009 (H1N1) (**c**), A/New York/384/2005 (H3N2) HA (**d**) or PMA/Ionomycin (PMA) and IFN-γ-producing cells in spleens from vaccinated mice were measured using an ELISpot assay. Data represent mean ± SEM spot forming cells (SFC) in each mouse (**a**,**b**) or pools from each group (**c**,**d**). Dotted lines (**a**–**d**) in each graph indicates average SFC count in the background no peptide wells. (**a**,**b**) Comparisons between sucrose alone or R-DOTAP groups was performed using Student’s *t*-test (unpaired-two tailed). ** *p* ≤ 0.05. Data shown are representative of two independent repeat studies.

**Figure 3 viruses-15-00432-f003:**
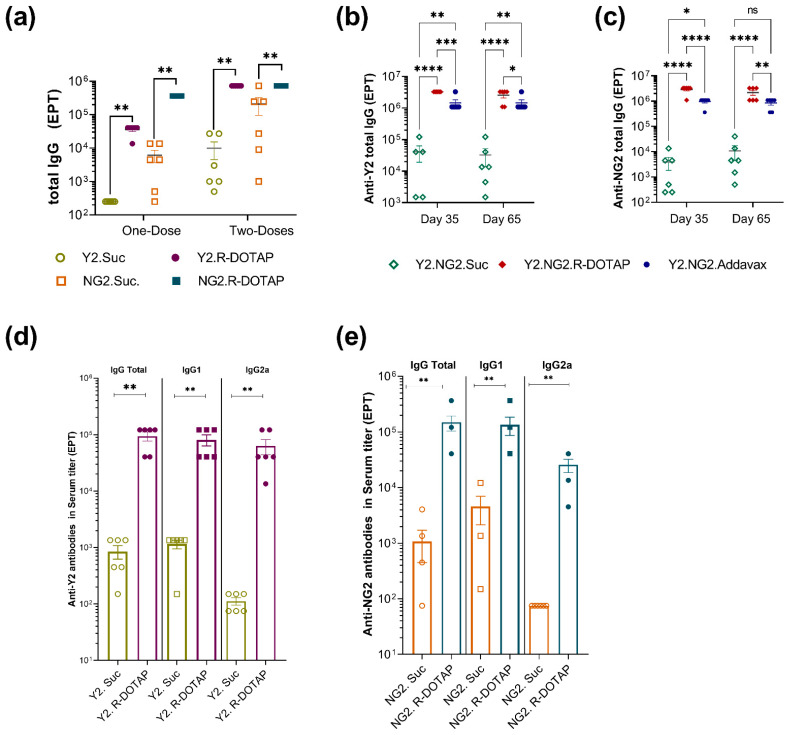
Antibody-mediated immune response induced by recombinant influenza protein formulated with R-DOTAP. Groups of BALB/cJ (*n* = 6–8) mice were immunized intramuscularly on day 0 and day 21 with two doses of monovalent (**a**) COBRA-Y2 (H1) (3 µg/dose), or COBRA NG2 (H3) (3 µg/dose) or bivalent vaccines (R-DOTAP-Y2NG2) containing COBRA-NG2 (H3) (3µg/dose), and COBRA-Y2 (H1) (3 µg/dose) (**b**–**e**), formulated with R-DOTAP nanoparticles, AddaVax or sucrose buffer (sucrose). (**a**) Serum samples obtained from vaccinated mice prior to second dose (one dose) on day 21 and 14 days after the second dose (two-doses) were measured for anti-COBRA-Y2 or Anti-COBRA-NG2 total IgG antibody titer by ELISA. Serum samples (**b**,**c**) obtained from vaccinated mice on day 35 (14 days after second dose) and day 65 (44 days after second dose) were measured for anti-COBRA-Y2, and anti-NG2 specific total IgG titers by ELISA. Serum samples (**d**,**e**) obtained from vaccinated mice on day 35 (14 days after the second dose) were measured for anti-Y2 and anti-NG2 total IgG, IgG1, and IgG2a antibody titer. Data represent (**a**,**d**,**e**) mean ± SEM end point titer (EPT) from each mouse. Data (**c**,**d**) represent reciprocal mean ± SEM of half-max titers from each mouse(**a**,**b**) Comparisons between sucrose alone or R-DOTAP groups was performed using Student’s *t*-test (unpaired-two tailed). **** *p* ≤ 0.0001, *** *p* ≤ 0.0003, ** *p* ≤ 0.05, * *p* ≤ 0.01, ^ns^
*p* ≥ 0.1. Data shown are representative of three independent repeat studies.

**Figure 4 viruses-15-00432-f004:**
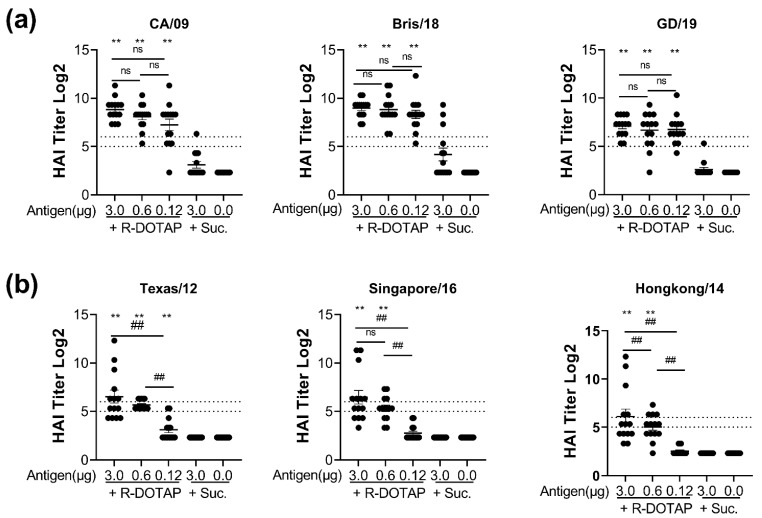
Bivalent COBRA antigens formulated with R-DOTAP induced robust neutralizing and broadly cross-reactive antibodies against H1N1 and H3N2 viruses. Groups of DBA/2J (*n* = 14) mice were immunized on day 0 and day 28 with two intramuscular doses of bivalent vaccines containing the indicated concentrations of COBRA-NG2 (H3), and COBRA-Y2 (H1) proteins formulated with R-DOTAP nanoparticles or sucrose buffer (no adj). Serum samples obtained from vaccinated mice 14 days after the second dose were assessed for functional antibodies by measuring HAI titers against H1N1 (**a**) viruses (A/California/09 (CA/09), A/Brisbane/18 (Bris/18), and Guangdong/19 (GD19)) or H3N2 (**b**) viruses (A/Texas/12, A/Singapore/16, Hongkong/14) viruses. Dotted lines represent HAI titers between 1:32 and1:64. Data represent mean ± SEM of HAI titers from each mouse. Comparisons between groups was performed using Kruskal–Wallis test in GraphPad Prism (9.0). ** *p* ≤ 0.05, significantly different between antigen+ sucrose and antigen + R-DOTAP. ## *p* ≤ 0.05 between groups compared. ns; not significant (*p* ≥ 0.05). Experiment was repeated in BALB/cJ mice with similar results.

**Figure 5 viruses-15-00432-f005:**
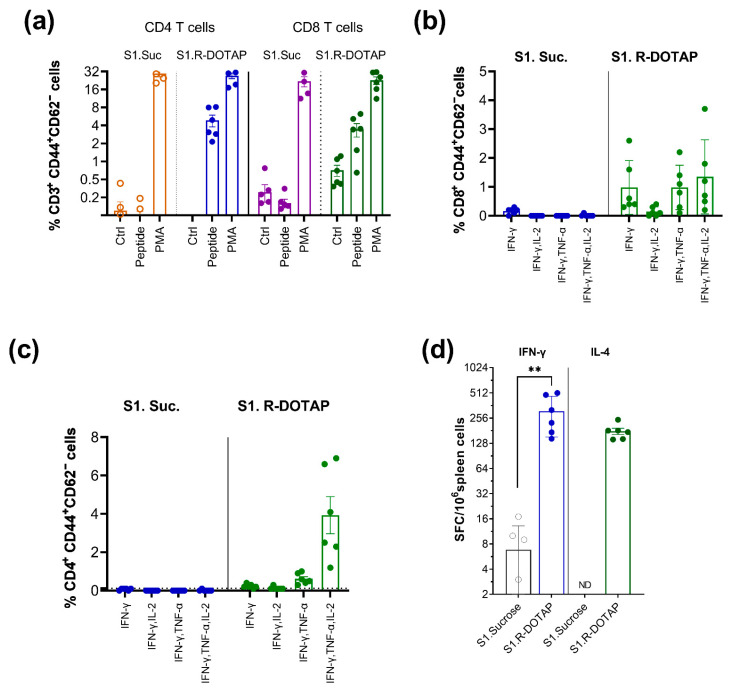
R-DOTAP-containing vaccine formulations induced polyfunctional T cells and long-term memory T-cells responses. Groups (**a**–**c**) of C57Bl/6J (*n* = 5) mice were subcutaneously injected on day 0 and day 21 with vaccine formulations containing 10 μg/dose spike S1 domain (S1), formulated with R-DOTAP nanoparticles (S1. R-DOTAP) or sucrose buffer (S1.Suc). Splenocytes isolated 7 days after second vaccine were stimulated overnight with spike S1 CD8 epitope (VL8) and CD4 epitope (VT15). Effector T cells (CD3^+^, CD44^+^, CD62L^−^) in spleen (**a**) producing IFN-γ in response to peptide stimulation (peptide) or PMA/ionomycin (PMA) or no stimulation (Ctrl) were measured cells using intracellular cytokine staining assay. (**b**,**c**) Polyfunctional CD8 T cells (**b**) and CD4 T cells (**c**) producing IFN-γ, IL-2 and TNF-alpha among gated (Appendix A) IFN-γ–producing effector cells (**a**) in spleen from vaccinated mice. Data represent mean ± SEM percentage of cytokine producing cells from each group. Groups (**d**) of BALB/cJ (*n* = 5) mice were immunized intramuscularly on day 0 and day 28 with 50 µL of vaccine formulations containing 10 μg/dose spike S1 domain formulated with R-DOTAP nanoparticles (S1.R-DOTAP) or sucrose buffer (S1. Sucrose). T–cell memory responses were measured 28 days after second dose (56 days total) by stimulation of spleen cells overnight with pools of overlapping peptides derived from SARS-CoV2 spike S1 domain in an IFN-γ or IL-4 ELISpot assay. Data represent background subtracted mean ± SEM spot-forming cells (SFC) in each mouse. Average background in unstimulated wells for this study had 31 ± 17 for IL-4 plates and 3.1 ± 1.5 for IFN-γ plates. (**d**) Comparisons between sucrose alone or R-DOTAP groups was performed using Student’s *t*-test (unpaired-two tailed). ** *p* ≤ 0.05. Data shown are representative of two independent repeat studies.

**Figure 6 viruses-15-00432-f006:**
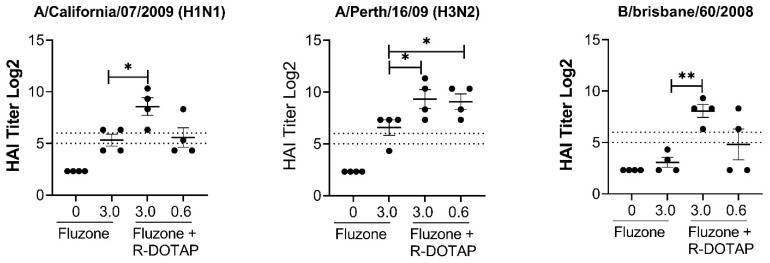
R-DOTAP significantly enhances the immunogenicity of Fluzone. Groups of C57BL/6J (*n* = 4) mice were subcutaneously vaccinated on day 0 and day 21 with indicated dose of Fluzone vaccine (2011–12 formula) mixed with R-DOTAP, or PBS buffer. Serum from vaccinated mice were obtained 14 days after boost and tested for neutralizing antibodies using a hemagglutination inhibition assay (HAI) against viruses A/Perth/16/2009 (H3N2), A/California/07/2009 (H1N1) and B Brisbane. Data represent mean ± SEM of HAI titers. Dotted lines represent HAI titers range of 1:32–1:64. Data represent mean ± SEM of HAI titers from each mouse. Comparisons between sucrose alone or R-DOTAP groups was performed using Student’s *t*-test (unpaired-two tailed). * *p* ≤ 0.1, ** *p* ≤ 0.05.

**Figure 7 viruses-15-00432-f007:**
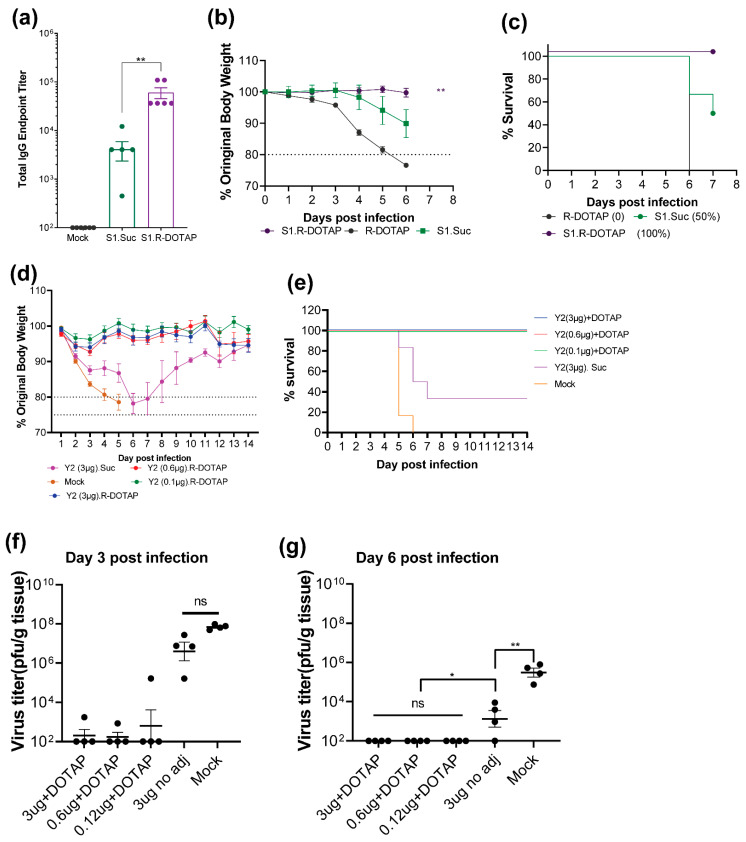
Mice vaccinated with Recombinant vaccines adjuvanted with R-DOTAP are protected from viral challenge. Groups (**a**–**c**) of K18hACE2 transgenic mice (*n* = 6) mice were intramuscularly injected on day 0 and day 28 with vaccine formulations containing recombinant SARS-CoV2 Spike S1 protein (10 µg/dose) in sucrose buffer (S1. Suc) or formulated with R-DOTAP nanoparticles (S1. R-DOTAP) or R-DOTAP nanoparticle alone (R-DOTAP). All vaccinated mice were challenged intranasally with SARS-CoV2 virus (2.5 × 10^4^ pfu/dose) (2019n-CoV/USA_WA1/2019 isolate) and weight loss and survival over an 8-day period was monitored. Mice that lost 20% of body weight were euthanized. Groups (**d**–**g**) of DBA-2 mice (*n* = 14) were intramuscularly injected on day 0 and day 28 with 50 µL of bivalent Y2NG2 (3–0.12 µg/dose) formulated with R-DOTAP nanoparticles (DOTAP) or sucrose buffer (suc). All vaccinated mice and unvaccinated mice (mock) were challenged intranasally with A/Brisbane/2/2018 (3.6 × 10^6^ pfu/dose) and weight loss and survival over two-week period was monitored. Mice that lost 20% of body weight during challenge were euthanized. Groups (**f**,**g**) of COBRA-vaccinated mice (*n* = 4) mice were euthanized on day 3 and day 6 and influenza viral titers in the lungs post challenge were measured using a plaque-forming assay. (**a**) Comparisons between sucrose alone or R-DOTAP groups was performed using Student’s *t*-test (unpaired-two tailed). (**b**) Two-way ANOVA was used to compare the weight change (until day 6) of S1. Suc, S1. R-DOTAP and R-DOTAP groups. (**f**,**g**) comparisons between groups were performed using One-Way ANOVA and Tukey multiple comparison test in GraphPad prism (9.0) ** *p* ≤ 0.05, * *p* ≤ 0.03, ns: *p* ≥ 0.1.

## Data Availability

The data that support the findings of this study are available from corresponding authors upon reasonable request, but restrictions apply to the availability of some proprietary information, which were used under license for the current study, and so are not publicly available.

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
