# Peer review of "Recombinant Protein Vaccines Formulated with Enantio-Specific Cationic Lipid R-DOTAP Induce Protective Cellular and Antibody-Mediated Immune Responses in Mice"

_viruses, 2023, doi:10.3390/v15020432_

Round 1

Reviewer 1 Report

This is an excellent study that points to the ability of R-DOTAP to enhance the immunogenicity of protein vaccines and their ability to promote protective immunity. The studies are comprehensive and for the most part the data support the conclusions.  The ability to generate neutralizing antibodies, as demonstrated by the in vitro assays and the lack of virus replication and detectable viral titers in the lung early after infection are impressive. Also, the use of multiple antigens and vaccine formuations speaks to the generality of the major conclusions. The authors are commended for their thoroughness and the comprehensive nature of their assays.

Several aspects of manuscript need to be improved with regard to clarity and controls. These are essential for interpretation of data and conclusions.

1. The authors and methods do not show the control values for the flow cytometry or Elispots, that would have been measured in the absence of peptide or with control peptides. These data are essential to provide to the reader and reviewer and are needed interpret the data. These should be provided in the legend of each figure in specific values or with data shown in additional panels for each mediator tested.

2. Flow cytometry is notoriously difficult to quantify and the gating schemes and primary data and negative control profiles values for the mediators are are essential to draw solid conclusions in Figure 6. These profiles need to be provided to the reader and reviewer as supplemental data and values for each mediator included in the legend to each Figure in the main manuscript.

3. The supplemental data shown in Figure S2b and 2c are interesting and striking. However, the legend  of the TEB is confusing. Were both the R-DOTAP and R-DOTAP-COBRA incubated with water only. As a lipid adjuvant, water alone would be a confusing solution. The legend needs to be clarified for both images.

Author Response

Response to Reviewer 1 Comments

Point 1: The authors and methods do not show the control values for the flow cytometry or Elispots, that would have been measured in the absence of peptide or with control peptides. These data are essential to provide to the reader and reviewer and are needed interpret the data. These should be provided in the legend of each Figure in specific values or with data shown in additional panels for each mediator tested.

Response 1: We appreciate reviewer’s suggestion for flow cytometry and Elispot assays. For all elispot assays, each mouse data was obtained from a triplicate well. As controls, we use unstimulated triplicate wells for each mouse i.e, cells with no peptide stimulation as negative controls, and PMA/Ionomycin wells for each mouse as positive control determining the health of the cells. For all our Elispot assays, the unstimulated wells are in the range 0-4 average spots. As suggested, these values have been added to the revised figure legends. For all flow cytometry assays, we use similar, unstimulated, and PMA-stimulated wells as negative and positive controls. As suggested by the reviewer, we have added a new graph to figure 5, and the gating scheme used to plot flow data as supplementary figure 3. 

Point 2: Flow cytometry is notoriously difficult to quantify and the gating schemes and primary data and negative control profiles values for the mediators are are essential to draw solid conclusions in Figure 6. These profiles need to be provided to the reader and reviewer as supplemental data and values for each mediator included in the legend to each Figure in the main manuscript.

Response 2: As requested by the reviewer, we have added a new graph to figure 5, and the gating scheme used to plot flow data as supplementary figure 3. 

Point 3: The supplemental data shown in Figure S2b and 2c are interesting and striking. However, the legend  of the TEB is confusing. Were both the R-DOTAP and R-DOTAP-COBRA incubated with water only. As a lipid adjuvant, water alone would be a confusing solution. The legend needs to be clarified for both images.

Response 3: As requested by the reviewer, we updated the Figure S2 b and 2C legends for clarity. 

Reviewer 2 Report

Intro

monophosphoryl lipid A would produce a strong cellular response like CpG or other TLR agonists. Also squalene adjuvants only produce a strong cellular response when something like an inactivated virus is present. AS03, MF59/Addavax, or IFA alone will not create a strong response without the inclusion of another PAMP. On the other hand agonists for TLR, NOD, STING, RIG-I and other PRRs will produce strong cellular responses. Please make sure your adjuvant discussion is as accurate as possible. Be critical of these cited work, if they are showing emulsions alone generating CD8 responses they might have endotoxin.

Like with the statement on adjuvants, this statement could be worded better “Vaccines against respiratory viruses, e.g. influenza or SARS CoV2, predominantly induce antibody responses and, for influenza, a neutralizing antibody titer…” there are some SARS CoV2 vaccines (like the Jansen one) that produce cellular responses. Also influenza vaccines like Flumist. Instead of ‘predominantly induce’, ‘often induce primarily’

It is unclear why in the introduction there is a discussion on lipid nanoparticles. The reason that LNPs work is because of ionizable lipids, not cationic lipids. Cationic lipids are not enough for efficient transfection. Now if there was a comparison that they lipid structures are FDA approved which paves the way for a lipid based adjuvant – maybe.

LN and HPV are not defined with first use. Acronyms should be defined with first use.

The figures are well laid out, but the captions are chaos to read. The explanations need to be simplified and presented in a straightforward and understandable manner. It is really hard to even discern how graphs A-C are different as the caption is written. I am still not sure what the different graphs are. With e, it is hard to tell the groups apart since the points are overlapping and one can’t see if the markers are filled or not because of that. Part of the confusion with A-C is that the same groups are also labeled differently on the X-axis. C looks like R-DOTAP was added as part of the recall, perhaps that was the case, but I don’t know because don’t understand the caption. It appears all captions are equally chaotic. Please keep in mind every experimental detail does not need to be in the captions. Please revise the figure captions in the manuscript.

For figure 7e, it would be helpful if the legend is in the same order as the survival since the lines overlap and are difficult to discern.

Discussion

“Adjuvants can be broadly categorized into Th1, Th2, Th17, and mixed Th1/Th2 or Th1/Th17 types based on the cytokines and antibody subclasses induced by the vaccine.” In this context, does Th17 need to be discussed? A focus on Th1 and Th2 is fine.

Author Response

Response to Reviewer 2 Comments

Point 1: monophosphoryl lipid A would produce a strong cellular response like CpG or other TLR agonists. Also squalene adjuvants only produce a strong cellular response when something like an inactivated virus is present. AS03, MF59/Addavax, or IFA alone will not create a strong response without the inclusion of another PAMP. On the other hand agonists for TLR, NOD, STING, RIG-I and other PRRs will produce strong cellular responses. Please make sure your adjuvant discussion is as accurate as possible. Be critical of these cited work, if they are showing emulsions alone generating CD8 responses they might have endotoxin.

 Response 1: We appreciate the reviewer’s suggestions and agree with his comments. We have made edits to our discussions to address these suggestions.

Point 2: Like with the statement on adjuvants, this statement could be worded better “Vaccines against respiratory viruses, e.g. influenza or SARS CoV2, predominantly induce antibody responses and, for influenza, a neutralizing antibody titer…” there are some SARS CoV2 vaccines (like the Jansen one) that produce cellular responses. Also influenza vaccines like Flumist. Instead of ‘predominantly induce’, ‘often induce primarily’.

Response 2: We appreciate the reviewer’s comments. We have made edits to the text to address these comments.

Point 3: It is unclear why in the introduction there is a discussion on lipid nanoparticles. The reason that LNPs work is because of ionizable lipids, not cationic lipids. Cationic lipids are not enough for efficient transfection. Now if there was a comparison that they lipid structures are FDA approved which paves the way for a lipid based adjuvant – maybe.

Response 3: We appreciate the reviewer’s comments. We added the lipid nanoparticle paragraph as a transition to our R-DOTAP introduction, which has slightly different properties from other lipid nanoparticles. R-DOTAP is a cationic lipid with two oleic acid side chains and a quaternary ammonium headgroup on one of the carbons of the lipid hydrocarbon backbone. This quaternary ammonium group is positively charged ordinarily at any neutral and acidic pH values. The molecule is not capable of attaining a negative charge and is PH-insensitive. We have shown in our earlier publications that this lipid has immune stimulatory activity in addition to efficient transfer of proteins. Hence we believe that this lipid can be a single-component adjuvant with dual roles as an efficient antigen carrier and an immune stimulant. We have demonstrated this using peptide vaccines, which are currently in phase II trials. We are excited and are currently exploring the possibility of its use as a prophylactic vaccine platform, and this paper demonstrates the proof- of concept. 

Point 4: LN and HPV are not defined with first use. Acronyms should be defined with first use.

Response 4: Thank you. We have made changes to reflect this.

Point 5: The figures are well laid out, but the captions are chaos to read. The explanations need to be simplified and presented in a straightforward and understandable manner. It is really hard to even discern how graphs A-C are different as the caption is written. I am still not sure what the different graphs are. With e, it is hard to tell the groups apart since the points are overlapping and one can’t see if the markers are filled or not because of that. Part of the confusion with A-C is that the same groups are also labeled differently on the X-axis. C looks like R-DOTAP was added as part of the recall, perhaps that was the case, but I don’t know because don’t understand the caption. It appears all captions are equally chaotic. Please keep in mind every experimental detail does not need to be in the captions. Please revise the figure captions in the manuscript. For figure 7e, it would be helpful if the legend is in the same order as the survival since the lines overlap and are difficult to discern..

Response 5: Thank you. We have revised all figure legends for clarity.

Point 6: “Adjuvants can be broadly categorized into Th1, Th2, Th17, and mixed Th1/Th2 or Th1/Th17 types based on the cytokines and antibody subclasses induced by the vaccine.” In this context, does Th17 need to be discussed? A focus on Th1 and Th2 is fine..

Response 6: Thank you. We have made changes to remove the Th17 discussion.

Round 2

Reviewer 1 Report

Thank you for responding to the issues raised in review. However,  for the most part, this reviewer was not able to locate the changes or description of needed controls that were requested in the original review. These are essential in interpretation of the data and the conclusions drawn from the authors. 

As discussed previously, the B cell assays and challenge data are compelling and well performed

1. First, for the flow cytometry data shown, was fluorescence minus one (FMO) used to set the gates? This is standard practice in flow cytometry. It was not clear from the images or the legend by what criteria the gates were drawn. This was not described in the methods or shown in the legends. 

2. The negative (no peptide) control was not shown or described for each cytokine in the ELISpot assays. This needs to be explicitly provided in the legend. 

3. The buffers used for the TEM in supplementary panels 2 were not provided.

Author Response

Point 1: First, for the flow cytometry data shown, was fluorescence minus one (FMO) used to set the gates? This is standard practice in flow cytometry. It was not clear from the images or the legend by what criteria the gates were drawn. This was not described in the methods or shown in the legends.

Response 1:  We have added additional text to the supplemental figure 3 legend and materials and methods to indicate the gating strategy. The gates were indeed set using separate FMO controls for IFN-g, TNF-a, and IL-2 as well as unstimulated controls (added as supplementary figure 3B). Similarly, the IFN-gamma gate was drawn by capturing both naïve and effector cell populations. We added these details to the supplementary figure legend.

Point 2: The negative (no peptide) control was not shown or described for each cytokine in the ELISpot assays. This needs to be explicitly provided in the legend.

Response 2:  We added a dotted line in each figure to indicate the average SFC observed in the experiment's negative ( no peptide controls).

Point 3: The buffers used for the TEM in supplementary panel two were not provided.

Response 3: Added text (lines 459-461) indicating the buffers used. As indicated TEM picture represents R-DOTAP in 280mM sucrose or Y2+NG2+R-DOTAP in 280mM sucrose. 

Round 3

Reviewer 1 Report

We appreciate the clarifications and added information needed to interpret the data.